# Discrete Geometry from Quantum Walks

**Fabrice Debbasch**

Sorbonne Université, Observatoire de Paris, Université PSL, CNRS, LERMA, F-75005 Paris, France; fabrice.debbasch@gmail.com

**Abstract:** A particular family of Discrete Time Quantum Walks (DTQWs) simulating fermion propagation in 2D curved space-time is revisited. Usual continuous covariant derivatives and spin-connections are generalized into discrete covariant derivatives along the lattice coordinates and discrete connections. The concepts of metrics and 2-beins are also extended to the discrete realm. Two slightly different Riemann curvatures are then defined on the space-time lattice as the curvatures of the discrete spin connection. These two curvatures are closely related and one of them tends at the continuous limit towards the usual, continuous Riemann curvature. A simple example is also worked out in full.

**Keywords:** discrete time quantum walks; discrete geometry; discrete Riemann curvature; discrete metric

---

## 1. Introduction

Discrete Time Quantum Walks (DTQWs) are unitary quantum automata. They have been first considered by Feynman [1] as tools to discretise path integrals for fermions, and later introduced in a more formal and systematic way in Aharonov [2] and Meyer [3]. DTQWs have been realized experimentally with a wide range of physical objects and setups [4–10], and are studied in a large variety of contexts, ranging from quantum optics [10] to quantum algorithmics [11,12], condensed matter physics [13–17], hydrodynamics [18] and biophysics [19,20].

It has been shown recently [21–30] that several DTQWs admit as continuous limit the dynamics of Dirac fermions coupled to arbitrary Yang–Mills gauge fields (including electromagnetic fields) and to arbitrary relativistic gravitational fields. In addition, a DTQW coupled to a uniform electric field has already been realized experimentally [31].

Quite remarkably, the DTQWs which converge towards fermions coupled to Yang–Mills fields also admit exact discrete gauge invariances and discrete field strength 'tensors'. These can be used e.g., to build new self-consistent discrete models of Dirac fermions interacting with Yang–Mills fields, where DTQWs are not only acted upon by gauge fields, but also act as sources to these fields. On the contrary, no exact discrete gauge invariance has been displayed for DTQWs which converge towards fermions coupled to gravitational fields, and no discrete field strength i.e., Riemann curvature has been defined either.

This article demonstrates how this gap can be filled. We focus on a certain family of DTQWs in discrete 2D space-time whose continuous limit coincides the dynamics of a (massless) Dirac fermion. We first show that these admit an exact discrete Lorentz gauge invariance and then present two alternate definitions of the Riemann curvature for these walks. More precisely, we define for each discrete walk discrete covariant derivatives in the direction of the grid coordinates. These derivatives generalise the usual covariant derivatives of differential geometry and allow the identification, not only of a discrete metric and a discrete 2-bein, but also of a discrete spin-connection defined on the lattice. The basic idea is then to define the Riemann curvature tensor of the space-time lattice (or of the DTQW)

---

as the curvature of the spin connection using as gauge group the set of Lorentz transformations acting on spinors [32,33]. It turns out that there are actually two ways of implementing this idea and we therefore introduce two different discrete Riemann curvatures on the space-time lattice. The first Riemann curvature $\rho^*$ depends on a (nearly arbitrary) reference connection while the second one $\rho^s$ does not. It turns out that the curvature $\rho^*$ of the DTQW essentially represents the difference between the curvature $\rho^s$ of the DTQW and the curvature $\rho^s$ of the reference connection. We also show that, in the continuous limit, the Riemann curvature tends towards the usual, continuous Riemann tensor. We finally compute the curvature $\rho^s$ on a simple example before discussing all results.

## 2. Results

### 2.1. Blueprint: The 2D Dirac Equation

The curved space-time Dirac equation is usually written in the form [34,35]

$$i\gamma^a e_a^\mu \mathcal{D}_\mu \Psi = m\Psi, \tag{1}$$

where $\Psi$ is a spinor, and $e_a^\mu$ are the $n$-bein coefficients, which we suppose to be symmetrical, i.e., $e_a^\mu = e_\mu^a$ (note that this relation makes sense because there are as many values of $\mu$ as there are values of $a$ and because it equates coefficients i.e., tensor components in a given basis, and not tensors). The $\gamma$'s are the so-called Dirac operators obeying the usual Clifford algebra, and

$$\mathcal{D}_\mu = \partial_\mu + \frac{1}{8}\omega_{\mu ab}\left[\gamma^a, \gamma^b\right]. \tag{2}$$

In 2D space-time, Greek and Latin indices above only take two values, conveniently denoted by 0 and 1. The spin-connection $\omega$ has thus only two independent components $\omega_{001} = -\omega_{010}$ and $\omega_{101} = -\omega_{110}$. The spinor Hilbert space is also two-dimensional and is equipped with the Hermitian product,

$$< \Psi(x^0, x^1), \Phi(x^0, x^1) > = \int_{x^1 \in \mathbb{R}} \Psi^*(x^0, x^1)\Phi(x^0, x^1)\mu(x^0, x^1)dx^1, \tag{3}$$

where $\mu = (-\det(g_{\mu\nu}))^{1/2}$ where $g_{\mu\nu}$ are the metric components built from the 2-bein, which can be defined by $g_{\mu\nu}e_a^\mu e_b^\nu = \eta_{ab}$ where $(\eta_{ab}) = \mathrm{diag}(1, -1)$ are the components of the 2D Minkovski metric in an orthonormal basis of the tangent space. We now choose an orthonormal basis $(b_-, b_+)$ in spinor space and represent an arbitrary spinor $\Psi$ by its two components $(\Psi^-, \Psi^+)$. We also choose the Dirac operators $\gamma^0$ and $\gamma^1$ to ensure that their matrix representations in this basis coincide respectively with $\sigma_x$ and $+i\sigma_y$ where

$$\sigma_x = \begin{pmatrix} 0 & 1 \\ 1 & 0 \end{pmatrix} \tag{4}$$

and

$$\sigma_y = \begin{pmatrix} 0 & -i \\ i & 0 \end{pmatrix} \tag{5}$$

are the first two Pauli matrices. The commutator then reads $\left[\gamma^0, \gamma^1\right] = 2\sigma_z = -2\mathrm{diag}(1, -1)$. Expanding the compact notation above, the Dirac therefore equation reads:

$$\begin{aligned}
(e_0^0 - e_1^0)\left(\partial_0\psi^- - \frac{\omega_{001}}{2}\psi^-\right) + (e_0^1 - e_1^1)\left(\partial_1\psi^- - \frac{\omega_{101}}{2}\psi^-\right) &= -im\psi^+, \\
(e_0^0 + e_1^0)\left(\partial_0\psi^+ + \frac{\omega_{001}}{2}\psi^+\right) + (e_0^1 + e_1^1)\left(\partial_1\psi^+ + \frac{\omega_{101}}{2}\psi^+\right) &= -im\psi^-.
\end{aligned} \tag{6}$$

The 2-bein, the metric and the two non-vanishing connection coefficients can then be practically read off directly from the Dirac equation. Taking the continuous limit of the QWs considered in this

article delivers this form of the Dirac equation [22]. In the next section, we will use discrete derivatives and put the QW equations in a form similar to (6) and thus identify in the discrete equations a 2-bein, a metric and a connection.

By definition, Lorentz transformations on spinors are generated by the commutator of the $\gamma's$. Thus, in 2D space-time, the Lorentz transform $\Psi(\Lambda)$ of a spinor $\Psi$ has components $\Psi^\pm(\Lambda) = \exp(\pm\Lambda)\Psi^\pm$ for an arbitrary, possibly space- and time-dependent $\Lambda$. In addition, the components of the spin connection transform according to $\omega_{\mu ab}(\Lambda) = \omega_{\mu ab} + \partial_\mu\Lambda$. It follows from this that $\mathcal{R}_{\mu\nu ab} = \partial_\mu\omega_{\nu ab} - \partial_\nu\omega_{\mu ab}$ is invariant under Lorentz transformation. This quantity is the $(\mu\nu ab)$-, so-called mixed component of the Riemann curvature tensor. The components $R_{\mu\nu\alpha\beta}$ of the Riemann curvature tensor on the coordinate basis $(\partial^\mu)$ are $R_{\mu\nu\alpha\beta} = \mathcal{R}_{\mu\nu ab}E^a_\alpha E^b_\beta$ where $(E^a_\alpha)$ are the coordinate basis components of the inverse 2-bein: $e^\mu_a E^b_\mu = \delta^a_b$. The Ricci tensor and the scalar curvature are defined from $R_{\mu\nu\alpha\beta}$ in the standard manner. Note that the expression of $\mathcal{R}_{\mu\nu ab}$ is linear in the connection because the Lorentz group is abelian in 2D space-time. In what follows, a discrete Riemann curvature tensor will be computed by implementing Lorentz transformations on the discrete equations and identifying an invariant quantity.

### 2.2. A Simple Two-Step Quantum Walk

We work with two-component wave-functions $\Psi$ defined in 2D discrete space-time where instants are labeled by $j \in \mathbb{N}$ and spatial positions are labeled by $p \in \mathbb{Z}$ and $\Psi_j = (\psi_{j,p})_{p\in\mathbb{Z}}$. We introduce a basis $(b_A) = (b_L, b_R)$ in Hilbert-space space and the components $\Psi^A = (\Psi^L, \Psi^R)$ of the arbitrary wave-function $\Psi$ in this basis. The Hilbert product is defined by $< \psi, \phi > = \sum_{A,j,p}(\psi^A)^*_{j,p}(\phi^A)_{j,p}$, which makes the basis $(b_A)$ orthonormal. Consider now the quantum walk $\Psi_{j+1} = U_j T\Psi_j$ where $T$ is the spin-dependent spatial translation operator defined by $(T\Psi_j)_{j,p} = (\psi^L_{j,p+1}, \psi^R_{j,p-1})^T$ and $U_j$ is an $SU(2)$ operator defined by

$$(U_j\Psi_j)_{j,p} = U(\theta_{j,p})\psi_{j,p}, \tag{7}$$

where

$$U(\theta) = \begin{pmatrix} -\cos\theta & i\sin\theta \\ -i\sin\theta & +\cos\theta \end{pmatrix}. \tag{8}$$

This article focuses on the two-step QW obtained by looking at the state of the original walk at only one in every two time steps, say the steps which correspond to even values of $j$ (this is sometimes called the stroboscopic approach).

Written in full, the discrete equations of the two-step QW read:

$$\begin{aligned} \psi^L_{j+2,p} &= c_{j+1,p}\left(c_{j,p+1}\psi^L_{j,p+2} - is_{j,p+1}\psi^R_{j,p}\right) + s_{j+1,p}\left(s_{j,p-1}\phi^L_{j,p} + ic_{j,p-1}\psi^R_{j,p-2}\right), \\ \psi^R_{j+2,p} &= s_{j+1,p}\left(ic_{j,p+1}\psi^L_{j,p+2} + s_{j,p+1}\psi^R_{j,p}\right) - c_{j+1,p}\left(is_{j,p-1}\psi^L_{j,p} - c_{j,p-1}\psi^R_{j,p-2}\right), \end{aligned} \tag{9}$$

where $c_{j,p} = \cos\theta_{j,p}$ and $s_{j,p} = \sin\theta_{j,p}$ As shown in [22], this two-step QW admits a continuous limit if $\theta$ admits one and this limit coincides with the Dirac equation in a curved 2D space-time where the spinor connection and curvature depend on the derivatives of $\theta$. The aim of this article is to show that the discrete equation can also be used to define a discrete metric, a discrete space-time connection and a discrete Riemann 'tensor' i.e., a full discrete geometry.

### 2.3. Covariant Discrete Derivatives

To define the geometry induced by this QW on the space-time lattice, it is necessary to change basis in the wave-function Hilbert space. The easiest way to do that is to write the equations of motion of the QW in an invariant, basis-independent manner by introducing covariant discrete derivatives in Hilbert space.

We start by defining the following simple, non covariant discrete derivatives:

$$(D_j f)_{jp} = \frac{1}{2}(f_{j+2,p} - f_{j,p}),$$
$$(D_p f)_{j,p} = \frac{1}{4}(f_{j,p+2} - f_{j,p-2}), \tag{10}$$
$$(D_{pp} f)_{j,p} = \frac{1}{4}(f_{j,p+2} + f_{j,p-2} - 2f_{j,p}),$$

where $f$ is an arbitrary $j$- and $p$-dependent quantity. These are discrete versions of the usual partial derivatives. Inverting the above equations delivers:

$$\begin{aligned}
f_{j+2,p} &= f_{j,p} + 2(D_j f)_{jp}, \\
f_{j,p+2} &= f_{j,p} + 2(D_p f)_{j,p} + 2(D_{pp} f)_{j,p}, \\
f_{j,p-2} &= f_{j,p} - 2(D_p f)_{j,p} + 2(D_{pp} f)_{j,p}.
\end{aligned} \tag{11}$$

The equation of motion of the QW can then be rewritten as:

$$(D_j \Psi^A)_{j,p} = (W_{j,p}\sigma_3)^A_B (D_p \Psi^B)_{j,p} + (1/2)(W_{j,p} + L_{j,p} - \mathbb{1})^A_B \Psi^B_{j,p} + (W_{j,p})^A_B (D_{pp} \Psi^B)_{j,p}, \tag{12}$$

where

$$(W^A_B)_{j,p} = \begin{pmatrix} c_{j+1,p}c_{j,p+1} & is_{j+1,p}c_{j,p-1} \\ is_{j+1,p}c_{j,p+1} & c_{j+1,p}c_{j,p-1} \end{pmatrix}, \tag{13}$$

$$(L^A_B)_{j,p} = \begin{pmatrix} s_{j+1,p}s_{j,p+1} & -ic_{j+1,p}s_{j,p+1} \\ -ic_{j+1,p}s_{j,p-1} & s_{j+1,p}s_{j,p+1} \end{pmatrix} \tag{14}$$

and $\sigma_3$ is the operator represented by the third Pauli matrix in the basis $(b_A)$ i.e., $\sigma_3$ is represented by the matrix $\text{diag}(1, -1)$ in the basis $(b_A)$.

Suppose now we change spin basis and rewrite (12) in a new, possibly $j$- and $p$-dependent local basis $b_\alpha = (b_-, b_+)$. We need to introduce the operator $r_{j,p}$ which transforms the original basis $b_A$ into the basis $b_\alpha$, $(b_\alpha)_{j,p} = (r_{j,p})^A_\alpha b_A$, and its inverse $r^{-1}_{j,p}$. Thus, $\psi_{jp} = \psi^A_{jp} b_A = \psi^A_{jp} ((r^{-1})^\alpha_A)_{jp} b_\alpha = \psi^\alpha_{jp} b_\alpha$ so that $\psi^\alpha_{jp} = ((r^{-1})^\alpha_A)_{jp} \psi^A_{jp}$ and $\psi^A_{jp} = (r^A_\alpha)_{jp} \psi^\alpha_{jp}$.

Let us now define covariant time- and space-derivatives, starting with derivation with respect to time. One has:

$$\begin{aligned}
(D_j \psi^A)_{j,p} &= \frac{1}{2}\left((r^A_\alpha)_{j+2,p}\psi^\alpha_{j+2,p} - (r^A_\alpha)_{j,p}\psi^\alpha_{j,p}\right) \\
&= \left((r^A_\alpha)_{j,p} + 2(D_j r^A_\alpha)_{j,p}\right)(D_j \psi^\alpha)_{j,p} + (D_j r^A_\alpha)_{j,p}\psi^\alpha_{j,p}.
\end{aligned} \tag{15}$$

This shows that $(D_j \psi^A)$ does not transforms as $\psi^A$ under a change of basis in Hilbert space, but this also suggests introducing a new, covariant time-derivative of the form

$$\left(\mathcal{D}_j(\mathcal{A})\psi^A\right)_{j,p} = \left(\mathcal{A}^1_{j,p}\right)^A_B (D_j \psi^B)_{j,p} + \left(\mathcal{A}^0_{j,p}\right)^A_B \psi^B_{j,p}, \tag{16}$$

where $(\mathcal{A}) = (\mathcal{A}^0, \mathcal{A}^1)$ is an arbitrary $j$- and $p$-dependent field. Using (15), one can write:

$$\left(\mathcal{D}_j(\mathcal{A})\psi^A\right)_{j,p} = (r^A_\alpha)_{j,p}\left(\mathcal{D}_j(\mathcal{A})\psi^\alpha\right)_{j,p}, \tag{17}$$

where

$$\left(\mathcal{D}_j(\mathcal{A})\psi^\alpha\right)_{j,p} = \left(\mathcal{A}^1_{j,p}\right)^\alpha_\beta (D_j \psi^\beta)_{j,p} + \left(\mathcal{A}^0_{j,p}\right)^\alpha_\beta \psi^\beta_{j,p}, \tag{18}$$

with

$$(\mathcal{A}^0)^\alpha_\beta = (r^{-1})^\alpha_A (\mathcal{A}^0)^A_B (r^B_\beta) + (r^{-1})^\alpha_A (\mathcal{A}^1)^A_B (D_j r^B_\beta) \tag{19}$$

and

$$(\mathcal{A}^1)^\alpha_\beta = (r^{-1})^\alpha_A (\mathcal{A}^1)^A_B (r^B_\gamma) \times \left( \delta^\gamma_\beta + 2(r^{-1})^\gamma_C (D_j r^C_\beta) \right), \tag{20}$$

and the time- and space-indices $j$ and $p$ have been omitted from the latest equations for readability purposes. Equation (17) proves that $\mathcal{D}_j(\mathcal{A})$ is a covariant time-derivative.

Space derivatives are slightly more complex. Using again (12), one can write:

$$
\begin{aligned}
(D_p \psi^A)_{j,p} &= \frac{1}{4} \left( (r^A_\alpha)_{j,p+2} \psi^\alpha_{j,p+2} - (r^A_\alpha)_{j,p-2} \psi^\alpha_{j,p-2} \right) \\
&= \left( (r^A_\alpha)_{j,p} + 2(D_{pp} r^A_\alpha)_{j,p} \right)(D_p \psi^\alpha)_{j,p} + (D_p r^A_\alpha)_{j,p} \left( \psi^\alpha_{j,p} + 2(D_{pp} \psi^\alpha)_{j,p} \right)
\end{aligned}
\tag{21}
$$

and

$$
\begin{aligned}
(D_{pp} \psi^A)_{j,p} &= \frac{1}{4} \left( (r^A_\alpha)_{j,p+2} \psi^\alpha_{j,p+2} + (r^A_\alpha)_{j,p-2} \psi^\alpha_{j,p-2} - 2(r^A_\alpha)_{j,p} \psi^\alpha_{j,p} \right) \\
&= \left( (r^A_\alpha)_{j,p} + 2(D_{pp} r^A_\alpha)_{j,p} \right)(D_{pp} \psi^\alpha)_{j,p} + 2(D_p r^A_\alpha)_{j,p}(D_p \psi^\alpha)_{j,p} + (D_{pp} r^A_\alpha)_{j,p} \psi^\alpha_{j,p}.
\end{aligned}
\tag{22}
$$

As before, this suggests defining a spatial covariant derivative by:

$$\left( \mathcal{D}_p(\mathcal{A}) \psi^A \right)_{j,p} = \left( \mathcal{A}^1_{j,p} \right)^A_B (D_p \psi^B)_{j,p} + \left( \mathcal{A}^0_{j,p} \right)^A_B \psi^B_{j,p} + \left( \mathcal{A}^2_{j,p} \right)^A_B (D_{pp} \psi^B)_{j,p} \tag{23}$$

where $(\mathcal{A}) = (\mathcal{A}^0, \mathcal{A}^1, \mathcal{A}^2)$ is an arbitrary $j$- and $p$-dependent field and the transformation laws for $\mathcal{A}$ reads

$$(\mathcal{A}^0)^\alpha_\beta = (r^{-1})^\alpha_A (\mathcal{A}^0)^A_B (r^B_\beta) + (r^{-1})^\alpha_A (\mathcal{A}^1)^A_B (D_p r^B_\beta) + (r^{-1})^\alpha_A (\mathcal{A}^2)^A_B (D_{pp} r^B_\beta), \tag{24}$$

$$(\mathcal{A}^1)^\alpha_\beta = (r^{-1})^\alpha_A (\mathcal{A}^1)^A_B (r^B_\gamma) \times \left( \delta^\gamma_\beta + 2(r^{-1})^\gamma_C (D_{pp} r^C_\beta) \right) + 2(r^{-1})^\alpha_A (\mathcal{A}^2)^A_B D_p r^B_\beta \tag{25}$$

and

$$(\mathcal{A}^2)^\alpha_\beta = (r^{-1})^\alpha_A (\mathcal{A}^2)^A_B (r^B_\gamma) \times \left( \delta^\gamma_\beta + 2(r^{-1})^\gamma_C (D_{pp} r^C_\beta) \right) + 2(r^{-1})^\alpha_A (\mathcal{A}^1)^A_B D_p r^B_\beta. \tag{26}$$

The equation of motion (12) of the QW can be rewritten in terms of covariant derivatives. We introduce a time-connection $\mathcal{A}$ and a space-connection $\mathcal{B}$, fixing only at this stage the values of their 1- and 2-components:

$$(\mathcal{A}^1)^A_B = \delta^A_B, \tag{27}$$

$$
\begin{aligned}
(\mathcal{B}^1)^A_B &= \delta^A_B, \\
(\mathcal{B}^2)^A_B &= (\sigma_3)^A_B.
\end{aligned}
\tag{28}
$$

We also introduce a 'mass' $\mathcal{M}$ and impose that

$$-i\mathcal{M}^A_B + (W\sigma_3 \mathcal{B}^0)^A_B - (\mathcal{A}^0)^A_B = (1/2)(W + L - \mathbb{1})^A_B, \tag{29}$$

thus ensuring that the equation of motion (12) can be written as:

$$\mathcal{D}_j(\mathcal{A}) \psi^A = (W\sigma_3)^A_B \mathcal{D}_p(\mathcal{B}) \psi^B - i\mathcal{M}^A_B \psi^B. \tag{30}$$

The 0-components of both connections and of the mass $\mathcal{M}$ will be specified in the next section. In addition, the status of $\mathcal{M}$ is discussed in the last section of this article.

Equation (30) is one step closer to the continuous Dirac equation that the original form of the equations of motion obeyed by the two-step walk. In particular, it shows that the time-connection $\mathcal{A}$

and the space-connection $\mathcal{B}$ are to be understood as two components of a single, space-time connection $(\mathcal{A}, \mathcal{B})$. This point of view will be adopted form here on.

The most important difference between (30) and (6) is that the operator $W\sigma_3$ is not diagonal in the basis $(b_A)$. Changing spin basis to make this operator diagonal is the goal of the next section.

### 2.4. Mass and Space-Time Connection

#### 2.4.1. Preliminary Gauge Change

Proceeding as in [22,23], we now change gauge i.e., spin basis by defining an operator $r$ which puts $W_{j,p}\sigma_3$ in diagonal form. The characteristic polynomial of $W_{j,p}\sigma_3$ reads

$$P_{j,p}(x) = x^2 + c_{j,p}\delta_{j,p}x - \pi_{j,p}, \tag{31}$$

where $\delta_{j,p} = c_{j,p-1} - c_{j,p+1}$ and $\pi_{j,p} = c_{j,p-1}c_{j,p+1}$. Let $(x_\alpha)_{j,p}$, $\alpha = +, -$ be the two (possibly complex) roots of $P_{j,p}$. From Equations (12) and (30), the eigenvalues $(x_\alpha)_{j,p}$ actually determine two local transport velocities. More precisely, these eigenvalues actually define a set of local 2-bein coefficients $(e_0^\mu, e_1^\nu)_{jp}$ (see above for details) on the space-time lattice. One finds $e_0^0 = 1$, $e_1^0 = 0$, $e_0^1 = (x_+ + x_-)/2$, $e_1^1 = (x_+ - x_-)/2$. This in turn defines the inverse metric 'components' on the space-time lattice $g^{00} = 1$, $g^{11} = x_+x_-$ and $g^{01} = (x_+ + x_-)/2$. The determinant of these components is $-\mu^2 = -(x_+ - x_-)^2/4$.

In the usual differential, and thus continuous geometry, the Greek indices on $n$-bein coefficients, (inverse) metric components, etc. refer to components on the so-called coordinate basis $(\partial_\mu) = (\partial_t, \partial_x)$. In the discrete case, the equivalent of the basis $(\partial_\mu)$ is clearly the set $(D_j, D_p)$ and we therefore define accordingly the 2-bein 'vectors' $e_0 = e_0^j D_j + e_0^p D_p$ and $e_1 = e_1^j D_j + e_1^p D_p$. The quantities $g^{\mu\nu}$ can be interpreted similarly as the components of the inverse metric $g^{jj}D_j \otimes D_j + 2g^{jp}D_j \otimes D_p + g^{pp}D_p \otimes D_p$. Changes of space-time coordinates can then be implemented in the spirit of [36]. We finally define the discrete inverse 2-bein by the usual relations $E_\mu^a e_b^\mu = \delta_b^a$ where $\delta_b^a$ is the Kronecker symbol.

We now recall that, in curved space-time, a spinor is normalized to unity, not with respect to the usual Lebesgue measure $d^2x$, but with respect to the metric-induced measure $\sqrt{(-\det g)}d^2x$ where $\det g$ stands for the determinant of the metric components. This means that the usual Hilbert product $<\psi, \phi> = \sum_{A,j,p}(\psi^A)_{j,p}^*(\phi^A)_{j,p}$, which makes the initial basis $b_A$ orthonormal, does not coincide with the natural Hilbertian product to be used in spinor space. We therefore define the new Hilbertian product by $<\psi, \phi>_s = \sum_{A,j,p}\mu_{j,p}(\psi^A)_{j,p}^*(\phi^A)_{j,p}$, a new basis $(b_\alpha)_{j,p}$ made of two eigenvectors of $W\sigma_3$ normalized with respect to $<\cdot>_s$ and we define $r_{j,p}$ as the operator which transforms the original basis $b_A$ into the basis $b_\alpha$.

#### 2.4.2. Choice of the Mass and Space-Time-Connection

Let us now specify the 0-components of the connections $\mathcal{A}$ and $\mathcal{B}$ as well as the mass $\mathcal{M}$. Equations (29), (27) and (28) lead to:

$$-i\mathcal{M}_\beta^\alpha + (W\sigma_3\mathcal{B}^0)_\beta^\alpha \quad - \quad (\mathcal{A}^0)_\beta^\alpha = \mathcal{N}_\beta^\alpha, \tag{32}$$

where

$$
\begin{aligned}
\mathcal{N}_\beta^\alpha &= (1/2)(W + L - \mathbb{1})_\beta^\alpha - (r^{-1})_A^\alpha(\mathcal{A}^1)_B^A D_j r_\beta^B + (W\sigma_3)_\gamma^\alpha(r^{-1})_A^\gamma\left((\mathcal{B}^1)_B^A D_p r_\beta^B + (\mathcal{B}^2)_B^A D_{pp} r_\beta^B\right) \\
&= (1/2)(W + L - \mathbb{1})_\beta^\alpha - (r^{-1})_A^\alpha D_j r_\beta^A + (W\sigma_3)_\gamma^\alpha(r^{-1})_A^\gamma\left(\delta_B^A D_p r_\beta^B + (\sigma_3)_B^A D_{pp} r_\beta^B\right).
\end{aligned}
\tag{33}
$$

We now define $-i\mathcal{M}_\beta^\alpha$ as the non-diagonal part of $\mathcal{N}_\beta^\alpha$. This fully specifies $\mathcal{M}$ in any basis of the Hilbert space and it also leads to

$$(W\sigma_3\mathcal{B}^0)_\beta^\alpha \quad - \quad (\mathcal{A}^0)_\beta^\alpha = \mathcal{O}_\beta^\alpha, \tag{34}$$

where $\mathcal{O}^\alpha_\beta$ is the diagonal part of $\mathcal{N}^\alpha_\beta$. Since $r$ was chosen to make $(W\sigma_3)^\alpha_\beta$ diagonal, this last equation makes it possible to choose both $(\mathcal{A}^0)^\alpha_\beta$ and $(\mathcal{B}^0)^\alpha_\beta$ diagonal, and (34) becomes a system of two equations for the four unknown $(\mathcal{A}^0)^-_-$, $(\mathcal{A}^0)^+_+$, $(\mathcal{B}^0)^-_-$, $(\mathcal{B}^0)^+_+$. In a generic situation, this system can be solved in a unique manner by imposing a couple of extra constraints on the unknown. We choose the same constraints as in the continuous case (see Section 2.1) i.e., $(\mathcal{A}^0)^-_- = -(\mathcal{A}^0)^+_+$ and $(\mathcal{B}^0)^-_- = -(\mathcal{B}^0)^+_+$, which make both $(\mathcal{A}^0)^\alpha_\beta$ and $(\mathcal{B}^0)^\alpha_\beta$ proportional to the third Pauli matrix $\sigma_z$.

### 2.5. Local Lorentz Transformations

Extending the definition of global Lorentz transformations for DTQWs proposed in [36], we now define the local Lorentz transform of the spinor $\Psi$ by $\psi^-_{j,p} \to \lambda_{j,p}\psi^-_{j,p}$ and $\psi^+_{j,p} \to \lambda^{-1}_{j,p}\psi^+_{j,p}$ for an arbitrary, real and non-vanishing field $\lambda$ defined on the 2D space-time lattice. Alternately, upon a Lorentz transformation, $\psi \to \exp(\Lambda\sigma_z)\psi$ where $\lambda = \exp(\Lambda)$ and $\sigma_z$ is the operator represented by the third Pauli matrix in the basis $(b_-, b_+)$, and we use the practical notation $\psi^\alpha(\Lambda) = \rho^\alpha_\beta(\Lambda)\psi^\beta$ where $\rho^\alpha_\beta(\Lambda) = \exp(\Lambda\sigma_z)$. Evidently, $(\rho^{-1})^\alpha_\beta(\Lambda) = \exp(-\Lambda\sigma_z)$.

Let us now compute the Lorentz transform of the DTQW equation of motion.

The mass $\mathcal{M}$ is anti-diagonal, so we write

$$\mathcal{M} = \begin{pmatrix} 0 & \mathcal{M}^-_+ \\ \mathcal{M}^+_- & 0 \end{pmatrix}, \tag{35}$$

which is not invariant under Lorentz transformation but becomes

$$\mathcal{M}(\Lambda) = \begin{pmatrix} 0 & e^{-2\Lambda}\mathcal{M}^-_+ \\ e^{+2\Lambda}\mathcal{M}^+_- & 0 \end{pmatrix}. \tag{36}$$

Note that the product $\mathcal{M}^-_+\mathcal{M}^+_-$, which can be interpreted as the squared mass of the walk, is invariant under Lorentz transformation.

The connection matrices also change under Lorentz transformation. Of particular interest are the diagonal parts of these connections because they obey a relatively simple transformation law. Indeed,

$$(\mathcal{A}^0)^-_-(\Lambda) = (\mathcal{A}^0)^-_- + (\mathcal{A}^1)^-_- \times \frac{1}{2}\left(\exp(2D_j\Lambda) - 1\right), \tag{37}$$

$$(\mathcal{A}^0)^+_+(\Lambda) = (\mathcal{A}^0)^+_+ + (\mathcal{A}^1)^+_+ \times \frac{1}{2}\left(\exp(-2D_j\Lambda) - 1\right), \tag{38}$$

$$\begin{aligned}(\mathcal{B}^0)^-_-(\Lambda) = {}& (\mathcal{B}^0)^-_- + \frac{1}{2}(\mathcal{B}^1)^-_- \exp(2D_{pp}\Lambda)\sinh(2D_p\Lambda) \\ & + (\mathcal{B}^2)^-_- \times \frac{1}{2}\left(\exp(2D_{pp}\Lambda)\cosh(2D_p\Lambda) - 1\right),\end{aligned} \tag{39}$$

$$\begin{aligned}(\mathcal{B}^0)^+_+(\Lambda) = {}& (\mathcal{B}^0)^+_+ + \frac{1}{2}(\mathcal{B}^1)^+_+ \exp(-2D_{pp}\Lambda)\sinh(-2D_p\Lambda) \\ & + (\mathcal{B}^2)^+_+ \times \frac{1}{2}\left(\exp(-2D_{pp}\Lambda)\cosh(-2D_p\Lambda) - 1\right),\end{aligned} \tag{40}$$

$$\begin{aligned}(\mathcal{B}^1)^-_-(\Lambda) = {}& (\mathcal{B}^1)^-_- \exp(2D_{pp}\Lambda)\cosh(2D_p\Lambda) \\ & + (\mathcal{B}^2)^-_- \exp(2D_{pp}\Lambda)\sinh(2D_p\Lambda),\end{aligned} \tag{41}$$

$$\begin{aligned}(\mathcal{B}^1)^+_+(\Lambda) = {}& (\mathcal{B}^1)^+_+ \exp(-2D_{pp}\Lambda)\cosh(-2D_p\Lambda) \\ & + (\mathcal{B}^2)^+_+ \exp(-2D_{pp}\Lambda)\sinh(-2D_p\Lambda).\end{aligned} \tag{42}$$

The first two equations lead to

$$\Delta\mathcal{A}^0(\Lambda) = (\mathcal{A}^1)^-_-(\mathcal{A}^1)^+_+ \sinh(2D_j\Lambda), \tag{43}$$

where

$$\Delta \mathcal{A}^0(\Lambda) = (\mathcal{A}^1)_+^+ \left( (\mathcal{A}^0)_-^-(\Lambda) - (\mathcal{A}^0)_-^- \right) - (\mathcal{A}^1)_-^- \left( (\mathcal{A}^0)_+^+(\Lambda) - (\mathcal{A}^0)_+^+ \right). \tag{44}$$

The following two equations lead to

$$
\begin{aligned}
\exp(+2D_{pp}\Lambda) &= 2 \frac{(\mathcal{B}^0)_-^-(\Lambda) - (\mathcal{B}^0)_-^- + (\mathcal{B}^2)_-^-/2}{(\mathcal{B}^1)_-^- \sinh(2D_p\Lambda) + (\mathcal{B}^2)_-^- \cosh(2D_p\Lambda)}, \\
\exp(-2D_{pp}\Lambda) &= 2 \frac{(\mathcal{B}^1)_+^+(\Lambda) - (\mathcal{B}^0)_+^+ + (\mathcal{B}^2)_+^+/2}{-(\mathcal{B}^1)_+^+ \sinh(2D_p\Lambda) + (\mathcal{B}^2)_+^+ \cosh(2D_p\Lambda)},
\end{aligned}
\tag{45}
$$

while the final two equations deliver

$$
\begin{aligned}
\exp(+2D_{pp}\Lambda) &= \frac{(\mathcal{B}^1)_-^-(\Lambda)}{(\mathcal{B}^1)_-^- \cosh(2D_p\Lambda) + (\mathcal{B}^2)_-^- \sinh(2D_p\Lambda)}, \\
\exp(-2D_{pp}\Lambda) &= \frac{(\mathcal{B}^1)_+^+(\Lambda)}{(\mathcal{B}^1)_+^+ \cosh(2D_p\Lambda) - (\mathcal{B}^2)_+^+ \sinh(2D_p\Lambda)}.
\end{aligned}
\tag{46}
$$

Equating both expressions of $\exp(\pm 2D_{pp}\Lambda)$ delivers

$$
\begin{aligned}
\tanh(+2D_p\Lambda) &= \frac{\mathcal{S}_-^-(\mathcal{B}(\lambda), \mathcal{B})}{\mathcal{C}_-^-(\mathcal{B}(\lambda), \mathcal{B})} = \mathcal{T}_-^-(\mathcal{B}(\lambda), \mathcal{B}), \\
\tanh(-2D_p\Lambda) &= \frac{\mathcal{S}_+^+(\mathcal{B}(\lambda), \mathcal{B})}{\mathcal{C}_+^+(\mathcal{B}(\lambda), \mathcal{B})} = \mathcal{T}_+^+(\mathcal{B}(\lambda), \mathcal{B}),
\end{aligned}
\tag{47}
$$

where

$$\mathcal{S}_-^-(\mathcal{B}(\lambda), \mathcal{B}) = -(\mathcal{B}^1)_-^- \left( (\mathcal{B}^0)_-^-(\Lambda) - (\mathcal{B}^0)_-^- + (\mathcal{B}^2)_-^-/2 \right) + (\mathcal{B}^2)_-^-(\mathcal{B}^1)_-^-(\Lambda)/2, \tag{48}$$

$$\mathcal{C}_-^-(\mathcal{B}(\lambda), \mathcal{B}) = +(\mathcal{B}^2)_-^- \left( (\mathcal{B}^0)_-^-(\Lambda) - (\mathcal{B}^0)_-^- + (\mathcal{B}^2)_-^-/2 \right) - (\mathcal{B}^1)_-^-(\mathcal{B}^1)_-^-(\Lambda)/2, \tag{49}$$

$$\mathcal{S}_+^+(\mathcal{B}(\lambda), \mathcal{B}) = +(\mathcal{B}^1)_+^+ \left( (\mathcal{B}^0)_+^+(\Lambda) - (\mathcal{B}^0)_+^+ + (\mathcal{B}^2)_+^+/2 \right) - (\mathcal{B}^2)_+^+(\mathcal{B}^1)_+^+(\Lambda)/2, \tag{50}$$

$$\mathcal{C}_+^+(\mathcal{B}(\lambda), \mathcal{B}) = +(\mathcal{B}^2)_+^+ \left( (\mathcal{B}^0)_+^+(\Lambda) - (\mathcal{B}^0)_+^+ + (\mathcal{B}^2)_+^+/2 \right) - (\mathcal{B}^1)_+^+(\mathcal{B}^1)_-^-(\Lambda)/2. \tag{51}$$

It is best to retain for $\tanh(+2D_p\Lambda)$ an expression which does not favour a set of components over the other. We therefore choose

$$\tanh(+2D_p\Lambda) = \frac{1}{2} \left( \mathcal{T}_-^-(\mathcal{B}(\Lambda), \mathcal{B}) - \mathcal{T}_+^+(\mathcal{B}(\Lambda), \mathcal{B}) \right) \tag{52}$$

as the final expression for $\tanh(+2D_p\Lambda)$.

*2.6. Riemann Curvature I*

Assuming that $(\mathcal{A}^1)_-^-(\mathcal{A}^1)_+^+$ does not vanish and inverting the functions sinh and tanh, Equations (43) and (52) can be rewritten under the form

$$
\begin{aligned}
D_j\Lambda &= L_j(\mathcal{A}(\Lambda), \mathcal{A}), \\
D_p\Lambda &= L_p(\mathcal{B}(\Lambda), \mathcal{B}).
\end{aligned}
\tag{53}
$$

The identity $[D_j, D_p] = 0$ then leads to

$$D_p L_j(\mathcal{A}(\Lambda), \mathcal{A}) - D_j L_p(\mathcal{B}(\Lambda), \mathcal{B}) = 0. \tag{54}$$

Introduce now a reference connection $(\mathcal{A}^*, \mathcal{B}^*)$, with the sole constraint that $L_j(\mathcal{A}^*, \mathcal{A})$ and $L_p(\mathcal{B}^*, \mathcal{B})$ are both defined, and write

$$
\begin{aligned}
L_j(\mathcal{A}(\Lambda), \mathcal{A}) &= L_j(\mathcal{A}_*, \mathcal{A}) + L_j^*(\mathcal{A}(\Lambda), \mathcal{A}), \\
L_p(\mathcal{B}(\Lambda), \mathcal{B}) &= L_p(\mathcal{B}_*, \mathcal{B}) + L_p^*(\mathcal{B}(\Lambda), \mathcal{B}).
\end{aligned}
\tag{55}
$$

Note that the identities $L_j(\mathcal{A}, \mathcal{A}) = L_p(\mathcal{B}, \mathcal{B}) = 0$ then imply

$$
\begin{aligned}
L_j^*(\mathcal{A}, \mathcal{A}) &= -L_j(\mathcal{A}^*, \mathcal{A}), \\
L_p^*(\mathcal{B}, \mathcal{B}) &= -L_p(\mathcal{B}^*, \mathcal{B}).
\end{aligned}
\tag{56}
$$

We then define the discrete Riemann curvature $\rho_{jp}^*(\Lambda)$ by

$$
\rho_{jp}^*(\Lambda) = + \left( D_p L_j^*(\mathcal{A}(\Lambda), \mathcal{A}) \right)_{j,p} - \left( D_j L_p^*(\mathcal{B}(\Lambda), \mathcal{B}) \right)_{j,p}.
\tag{57}
$$

By (56),

$$
\rho_{jp}^*(0) = - \left( D_p L_j(\mathcal{A}^*, \mathcal{A}) \right)_{j,p} + \left( D_j L_p(\mathcal{B}^*, \mathcal{B}) \right)_{j,p},
\tag{58}
$$

which represents the discrete Riemann curvature $\rho^*$ of the connection $(\mathcal{A}, \mathcal{B})$ i.e., the curvature $\rho^*$ of the DTQW.

## 2.7. Riemann Curvature II

Suppose now that one is interested in a curvature which caracterizes only how the connection coefficients change under Lorentz transformations which vary slowly in time and space i.e., for which $D_j\Lambda$, $D_p\Lambda$ and $D_{pp}\Lambda$ are all much smaller than unity. At the continuous limit, all Lorentz transformations are automatically slowly varying in both time and space because the time and space coordinates $t$ and $x$ are related to $j$ and $p$ by $t_j = \epsilon j$ and $x_p = \epsilon p$, where $\epsilon$ is an infinitesimal [22,23], so that $D_p \sim \epsilon \partial_x$ and $D_{pp} \sim \epsilon^2 \partial_{xx}$. However, slowly varying Lorentz transformations can also be considered outside the continuous limit (see the example in the next section).

The limit case of Lorentz transformations varying slowly in space is actually singular. Indeed, in the general case, Equations (37)–(42) relate the two independent variations $\mathcal{B}^0(\Lambda) - \mathcal{B}^0$ and $\mathcal{B}^1(\Lambda) - \mathcal{B}^1$ to the two independent discrete derivatives $D_p\Lambda$ and $D_{pp}\Lambda$. Inverting these equations thus delivers $D_p\Lambda$ in terms of the two independent variables $\mathcal{B}^0(\Lambda) - \mathcal{B}^0$ and $\mathcal{B}^1(\Lambda) - \mathcal{B}^1$. To study the limit case of slowly varying Lorentz transformations, suppose $D_p\Lambda = O(\epsilon)$ and $D_{pp}\Lambda = O(\epsilon^\alpha)$ with $\alpha > 1$. Equations (37)–(42) then read:

$$
(\mathcal{B}^0)_-^-(\Lambda) = (\mathcal{B}^0)_-^- + (\mathcal{B}^1)_-^- D_p\Lambda + o(\epsilon),
\tag{59}
$$

$$
(\mathcal{B}^0)_+^+(\Lambda) = (\mathcal{B}^0)_+^+ - (\mathcal{B}^1)_+^+ D_p\Lambda + o(\epsilon),
\tag{60}
$$

$$
(\mathcal{B}^1)_-^-(\Lambda) = (\mathcal{B}^1)_-^- + 2(\mathcal{B}^2)_-^- D_p\Lambda + o(\epsilon),
\tag{61}
$$

$$
(\mathcal{B}^1)_+^+(\Lambda) = (\mathcal{B}^1)_+^+ - 2(\mathcal{B}^2)_+^+ D_p\Lambda + o(\epsilon).
\tag{62}
$$

At the first order in $\epsilon$, $D_{pp}\Lambda$ vanishes from the equations so both variations $\bar{\mathcal{B}}^0(\Lambda) = \mathcal{B}^0(\Lambda) - \mathcal{B}^0$ and $\bar{\mathcal{B}}^1(\Lambda) = \mathcal{B}^1(\Lambda) - \mathcal{B}^1$ depend on the single variable $D_p\Lambda$ and they are therefore not independent. Indeed, $\bar{\mathcal{B}}^1(\Lambda) = 2\mathcal{B}^2 \bar{\mathcal{B}}^0(\Lambda)$. In this limit, the general problem, which depends on two variables, thus degenerates into a single variable problem, thus making the limit singular. To define curvature,

one then needs only one of the two variations and it is natural to retain $\bar{\mathcal{B}}^0(\Lambda)$. The equation for $D_p\Lambda$ then reads

$$D_p\Lambda \approx \frac{1}{2}\left(\frac{(\bar{\mathcal{B}}^0)^-_-(\Lambda)}{(\mathcal{B}^1)^-_-} - \frac{(\bar{\mathcal{B}}^0)^+_+(\Lambda)}{(\mathcal{B}^1)^+_+}\right) \tag{63}$$

and the equation for $D_j\Lambda$ becomes similarly

$$D_j\Lambda \approx \frac{1}{2}\left(\frac{(\bar{\mathcal{A}}^0)^-_-(\Lambda)}{(\mathcal{A}^1)^-_-} - \frac{(\bar{\mathcal{A}}^0)^+_+(\Lambda)}{(\mathcal{A}^1)^+_+}\right), \tag{64}$$

where $\bar{\mathcal{A}}(\Lambda) = \mathcal{A}(\Lambda) - \mathcal{A}$.

From this choice and the identity $[D_j, D_p] = 0$ follows

$$\begin{aligned}
0 &= \frac{1}{2}D_j\left(\frac{(\bar{\mathcal{B}}^0)^-_-(\Lambda)}{(\mathcal{B}^1)^-_-} - \frac{(\bar{\mathcal{B}}^0)^+_+(\Lambda)}{(\mathcal{B}^1)^+_+}\right)\\
&\quad - \frac{1}{2}D_p\left(\frac{(\bar{\mathcal{A}}^0)^-_-(\Lambda)}{(\mathcal{A}^1)^-_-} - \frac{(\bar{\mathcal{A}}^0)^+_+(\Lambda)}{(\mathcal{A}^1)^+_+}\right),
\end{aligned} \tag{65}$$

where the $s$ index stands for 'slow'. The 'slow' discrete Riemann curvature tensor $\rho^s_{jp}(\Lambda)$ of a connection is then defined by:

$$\rho^s_{jp}(\Lambda) = \frac{1}{2}D_j\left(\frac{(\mathcal{B}^0(\Lambda))^-_-}{(\mathcal{B}^1)^-_-} - \frac{(\mathcal{B}^0(\Lambda))^+_+}{(\mathcal{B}^1)^+_+}\right) - \frac{1}{2}D_p\left(\frac{(\mathcal{A}^0(\Lambda))^-_-}{(\mathcal{A}^1)^-_-} - \frac{(\mathcal{A}^0(\Lambda))^+_+}{(\mathcal{A}^1)^+_+}\right), \tag{66}$$

where the index 's' stands for slow, ensuring that $\rho^s_{jp}(\Lambda) = \rho^s_{jp}(0)$. In addition, the Riemann of the DTQW is defined as $\rho^s_{jp}(0)$.

### 2.8. Relation between the Two Riemann Curvatures

Let us now investigate how this second discrete Riemann tensor is related to the first one introduced in the previous section. To do so, suppose that both connections $(\mathcal{A}^*, \mathcal{B}^*)$ and $(\mathcal{A}(\Lambda), \mathcal{B}(\Lambda))$ are close to $(\mathcal{A}, \mathcal{B})$, in the sense that their coefficients in the basis $(b_\alpha)$ are close to those of $(\mathcal{A}, \mathcal{B})$. This implies in particular that $D_j\Lambda$, $D_p\Lambda$ and $D_{pp}\Lambda$ are small (see Equations (37)–(42)) i.e., that $\Lambda$ is slowly varying in time and space. To simplify the discussion, we also suppose that there exist a $\Lambda^*$ such that $(\mathcal{A}^*, \mathcal{B}^*) = (\mathcal{A}(\Lambda^*), \mathcal{B}(\Lambda^*))$, and $\Lambda^*$ is then also slowly varying in time and space. We now convert $L_j$ and $L_p$ into a function $\bar{L}_j$ of $(\bar{\mathcal{A}}(\Lambda), \mathcal{A})$ and a function $\bar{L}_p$ of $(\bar{\mathcal{B}}(\Lambda), \mathcal{B})$ and expand these two newly introduced functions in their first variable at first order around 0. This leads to:

$$\begin{aligned}
\bar{L}_j(\bar{\mathcal{A}}(\Lambda), \mathcal{A}) &\approx \bar{\mathcal{A}}(\Lambda)\left(\frac{\partial\bar{L}_j}{\partial\bar{\mathcal{A}}}\right)_{(0,\mathcal{A})},\\
\bar{L}_j(\bar{\mathcal{A}}(\Lambda^*), \mathcal{A}) &\approx \bar{\mathcal{A}}(\Lambda^*)\left(\frac{\partial\bar{L}_j}{\partial\bar{\mathcal{A}}}\right)_{(0,\mathcal{A})},\\
\bar{L}_p(\bar{\mathcal{B}}(\Lambda), \mathcal{B}) &\approx \bar{\mathcal{B}}(\Lambda)\left(\frac{\partial\bar{L}_p}{\partial\bar{\mathcal{B}}}\right)_{(0,\mathcal{B})},\\
\bar{L}_p(\bar{\mathcal{B}}(\Lambda^*), \mathcal{B}) &\approx \bar{\mathcal{B}}(\Lambda^*)\left(\frac{\partial\bar{L}_p}{\partial\bar{\mathcal{B}}}\right)_{(0,\mathcal{B})}.
\end{aligned} \tag{67}$$

From this follows that

$$
\begin{aligned}
\bar{L}_j^*(\bar{\mathcal{A}}(\Lambda), \mathcal{A}) &\approx (\mathcal{A}(\Lambda) - \mathcal{A}(\Lambda^*)) \left(\frac{\partial \bar{L}_j}{\partial \bar{\mathcal{A}}}\right)_{(0, \mathcal{A})}, \\
\bar{L}_p^*(\bar{\mathcal{B}}(\Lambda), \mathcal{B}) &\approx (\mathcal{B}(\Lambda) - \mathcal{B}(\Lambda^*)) \left(\frac{\partial \bar{L}_p}{\partial \bar{\mathcal{B}}}\right)_{(0, \mathcal{B})},
\end{aligned}
\tag{68}
$$

which leads to

$$
\rho_{jp}^*(\Lambda) \approx \bar{\rho}_{jp}(\Lambda) - \bar{\rho}_{jp}(\Lambda^*),
\tag{69}
$$

where

$$
\bar{\rho}_{jp}(\Lambda) = D_p \left( \bar{\mathcal{A}}(\Lambda) \left(\frac{\partial \bar{L}_j}{\partial \bar{\mathcal{A}}}\right)_{(0, \mathcal{A})} \right) - D_j \left( \bar{\mathcal{B}}(\Lambda) \left(\frac{\partial \bar{L}_p}{\partial \bar{\mathcal{B}}}\right)_{(0, \mathcal{B})} \right).
\tag{70}
$$

Using again (67), this becomes

$$
\bar{\rho}_{jp}(\Lambda) = D_p \left( \bar{L}_j(\bar{\mathcal{A}}(\Lambda), \mathcal{A}) \right) - D_j \left( \bar{L}_p(\bar{\mathcal{B}}(\Lambda), \mathcal{B}) \right).
\tag{71}
$$

Now, by definition, $\bar{L}_j$ represents $D_j \Lambda$ and $\bar{L}_p$ represents $D_p \Lambda$. Since we are considering $\Lambda$'s which vary slowly in time and space, Equations (63) and (64) are valid. Thus,

$$
\rho_{jp}^*(\Lambda) \approx \rho_{jp}^s(\Lambda) - \rho_{jp}^s(\Lambda^*).
\tag{72}
$$

In particular, $\rho_{jp}^*(0) = \rho_{jp}^s(0) - \rho_{jp}^s(\Lambda^*)$, which links the two Riemann curvatures $\rho^*$ and $\rho^s$ of the space-time lattice.

### 2.9. Continuous Limit

Let us now discuss the continuous limit of $\rho^s$. The continuous limit addresses situations where the operator $\mathcal{U}$ and the wave-function of the walk vary on time- and space-scale much larger than the grid cell. The physical time $t$ and spatial coordinate $x$ along the grid are related to $j$ and $p$ by $t_j = \epsilon j$ and $x_p = \epsilon p$ where $\epsilon$ is an infinitesimal. It has been shown in [22,23] that the continuous limit of the 2-step walk then coincides with the Dirac equation in a curved space-time with metric $(g_{\mu\nu}) = \mathrm{diag}(1, \cos^{-2}\theta)$. In particular, the matrices representing $\mathcal{A}^1$ and $\mathcal{B}^1$ in the basis $(b_\alpha)$ then tend towards unity while the matrix representing $\mathcal{A}^0$ tends towards $-(\omega_{001}/2) \times \mathbb{1}$ and $\mathcal{B}^0$ tends towards $-(\omega_{101}/2) \times \mathbb{1}$. The discrete Riemann curvature then tends towards $\epsilon^2/2 \times R_{\mu\nu ab}$, where $R_{\mu\nu ab}$ is the mixed component of the usual Riemann curvature tensor to $\mu = 0, \nu = 1, a = 0, b = 1$. This component contains all the information one needs about the Riemann tensor because this tensor, in 2D space-times, has only one independent component. The $1/2$ in the multiplicative factor comes form the fact that the zeroth components of the discrete connection tend towards $\omega/2$ (as opposed to $\omega$). The $\epsilon^2$ factor comes from the fact that curvatures are obtained by taking second discrete or continuous derivatives and that the above relation between $(j, p)$ and $(t, x)$ implies $D_j = \epsilon \partial_t$ and $D_p = \epsilon \partial_x$. Finally, the components $R_{\mu\nu 01}$ of the continuous Riemann curvature tensor on the coordinate basis $(\partial_\mu)$ can be recovered by taking the continuous limit of $E_\mu^a E_\nu^b \rho^s$ where $(E_\mu^a)$ is the discrete inverse 2-bein.

### 2.10. Example

The continuous limit of the walks studied in this article corresponds to the propagation of a Dirac spinor in a space-time metric of the form $ds^2 = dt^2 - a^2(t, x) dx^2$ where $t$ and $x$ are the continuous coordinates corresponding to $j$ and $p$ and $a(t) = 1/(\cos\theta)$. Fixing these coordinates i.e., retaining this form for the metric, the simplest space-times with non vanishing curvature are realized by choosing the function $a$ independent of $x$. We now therefore choose an angle $\theta$ which depends only on $j$ and

proceed to compute, as an example, the first of the curvatures defined above. Since nothing depends on the spatial position, all quantities are now indexed by $j$ only.

For such walks, the operators $W$ and $L$ take the simpler form

$$(W_B^A)_j = c_j \begin{pmatrix} c_{j+1} & i s_{j+1} \\ i s_{j+1} & c_{j+1} \end{pmatrix}, \tag{73}$$

$$(L_B^A)_j = s_j \begin{pmatrix} s_{j+1} & -i c_{j+1} \\ -i c_{j+1} & s_{j+1} \end{pmatrix} \tag{74}$$

and

$$((W+L)_B^A)_j = \begin{pmatrix} \cos(\Delta\theta_j) & i \sin(\Delta\theta_j) \\ i \sin(\Delta\theta_j) & \cos(\Delta\theta_j) \end{pmatrix} \tag{75}$$

with $\Delta\theta_j = \theta_{j+1} - \theta_j$.

A simple computation leads to $(x_\pm)_j = \pm \mid c_j \mid$. These values of $(x_\pm)_j$ lead to $(g^{jj})_j = 1$, $(g^{pp})_j = -c_j^2$ and $(g^{jp})_j = 0$. If $\theta_j \neq \pi/2$, the components of the discrete metric itself read $(g_{jj})_j = 1$, $(g_{pp})_j = -c_j^{-2}$ and $(g_{jp})_j = 0$. In addition, $-\mu^2 = -c_j^2$.

We now retain (assuming $c_j \neq 0$)

$$\begin{aligned} (b_-)_j &= \mid c_j \mid^{-1/2} \left( i\sigma_j b_L + \kappa_j b_R \right), \\ (b_+)_j &= \mid c_j \mid^{-1/2} \left( \kappa_j b_L + i\sigma_j b_R \right), \end{aligned} \tag{76}$$

where $\kappa_j = \cos(\theta_{j+1}/2)$ and $\sigma_j = \sin(\theta_{j+1}/2)$. The matrix $(r_\alpha^A)_j$ can be read off these equations:

$$(r_\alpha^A)_j = \mid c_j \mid^{-1/2} \begin{pmatrix} i\sigma_j & \kappa_j \\ \kappa_j & i\sigma_j \end{pmatrix} \tag{77}$$

and its inverse reads:

$$((r^{-1})_A^\alpha)_j = \mid c_j \mid^{+1/2} \begin{pmatrix} -i\sigma_j & \kappa_j \\ \kappa_j & -i\sigma_j \end{pmatrix}. \tag{78}$$

The components of $W + L$ are not modified by the change of basis i.e.,

$$((W+L)_\beta^\alpha)_j = \begin{pmatrix} \cos(\Delta\theta_j) & i \sin(\Delta\theta_j) \\ i \sin(\Delta\theta_j) & \cos(\Delta\theta_j) \end{pmatrix} \tag{79}$$

and a direct computation delivers

$$(\mathcal{M}_\beta^\alpha) = \begin{pmatrix} 0 & \bar{\mathcal{M}} \\ \bar{\mathcal{M}} & 0 \end{pmatrix} \tag{80}$$

with

$$\bar{\mathcal{M}} = -\frac{1}{2}\sin(\Delta\theta) + \mid c \mid^{+1/2} \left( \kappa D_j(\mid c \mid^{-1/2} \sigma) - \sigma D_j(\mid c \mid^{-1/2} \kappa) \right), \tag{81}$$

where the index $j$ tracing the time-dependence of all quantities has been suppressed for readability purposes.

Since all angles depend only on $j$, only the connection $\mathcal{B}$ enters the curvature. One finds that

$$((\mathcal{B}^1)_\beta^\alpha) = \mathbb{1}, \tag{82}$$

$$(\mathcal{B}^0)_-^- = - \mid c \mid^{-1/2} \left( \kappa D_j(\mid c \mid^{-1/2} \kappa) + \sigma D_j(\mid c \mid^{-1/2} \sigma) \right) + \frac{1}{2\mid c \mid}(\cos\Delta\theta - 1), \tag{83}$$

while $(\mathcal{B}^0)_+^+ = -(\mathcal{B}^0)_-^-$ and

$$((\mathcal{B}^2)_\beta^\alpha)_j = \begin{pmatrix} -\cos(\theta_{j+1}) & -i\sin(\theta_{j+1}) \\ i\sin(\theta_{j+1}) & \cos(\theta_{j+1}) \end{pmatrix}. \tag{84}$$

This leads to $\rho_j^s = D_j(\mathcal{B}^0)_-^-$ with $(\mathcal{B}^0)_-^-$ given by Equation (83).

## 3. Conclusions

We have revisited a particular family of DTQWs whose continuous limit coincides with the 2D curved space-time Dirac dynamics written in synchronous coordinates. We have defined discrete covariant derivatives of the spinor wave-function along the grid coordinates, thus introducing discrete spin-connections and also generalised the notions of metric and 2-bein to the discrete lattice. We have then defined two different discrete curvatures from the transformation properties of the discrete spin-connections under Lorentz transformations. Both curvatures are closely related and one of them coincides, in the continuous limit, with the usual Riemann curvature from differential geometry. We have finally computed this discrete Riemann curvature on a particularly simple example.

Let us now comment on these results. In an arbitrary space-time, the most complete characterization of curvature is given the Riemann tensor. This tensor is usually computed from the space-time connection, but it can also be obtained from spinor connection [32,33]. The definition and computation of discrete curvature presented in this article thus start with a definition of discrete spinor connections for DTQWs, which is itself based upon the definition of discrete first and second discrete partial derivatives with respect to the grid coordinates. In the discrete case, spinor connections have a richer structure than in the continuous case because they contain more coefficients. In 2D space-time, a continuous spinor connection is fully defined by two coefficients, whereas one needs five coefficients to fully define a discrete spinor connection. These five coefficients can be partitioned into two sets, one of two coefficients pertaining to discrete covariant derivatives with respect to the discrete time index $j$, and one of three coefficients pertaining to covariant derivatives with respect to the discrete space index $p$. Note that these two sets only mix if one performs discrete Lorentz transformations in space-time, and these have not been considered in this article, where only Lorentz transformations in spinor space are carried out. We have therefore chosen, for readability purposes, to use a different letter for each set of coefficients ($\mathcal{A}$ defines discrete covariant time-derivatives and $\mathcal{B}$ defines discrete covariant space-derivatives). In addition, the discrete space-time connection is thus represented by $(\mathcal{A}, \mathcal{B})$.

The computation of the Riemann tensor as the curvature of the spin connection coefficients using as gauge group the set of Lorentz transformations in spinor space does not deliver the usual space-time components of the tensor, but the so-called mixed components $R_{\mu\nu ab}$, from which the usual space-time components can be recovered through partial contraction with the inverse $n$-bein coefficients. This applies both to the continuous and the discrete case. In 2D, there is only one independent component to the usual continuous Riemann tensor, and the discrete one also has only one independent component.

We have proposed two different definitions of Riemann curvature for the DTQWs considered in this article. If one is mainly interested in quantum simulation of conventional continuous physics, the second definition, which makes use of slowly varying Lorentz transformations, is clearly the one of choice, if only because its continuous limit gives back the usual Riemann curvature of differential geometry. However, DTQWs are interesting in other contexts, for example in quantum computing and quantum algorithms, where the continuous limit is not necessarily of particular importance. It is therefore useful to develop, for these contexts, a very general notion of curvature which is not linked to what happens at the continuous limit or for slowly varying Lorentz transformations. This is why we have offered our first definition of Riemann curvature. For technicality reasons, the first definition makes it necessary to introduce a reference connection and the obtained Riemann curvature

thus depends on this reference connection. In essence, the reference connection is the connection for which the first Riemann curvature vanishes. The easiest way to see this is to go back to Equation (72). This states that the first curvature of a connection is the difference between the second curvature of that connection and the second curvature of the reference connection. There is no canonical choice for the reference connection i.e., the reference connection must be chosen on case by case basis, according to the context and interests of the computation.

The whole approach developed in this article is close in spirit to work which has been done in the last fifteen years, where classical Markov chains are used to define Ricci curvatures of graphs [37–39]. Indeed, a Markov chain is essentially a discrete diffusion. It therefore defines a Laplace operator on the discrete structure where it lives and, thus, a Ricci curvature. Similarly, a DTQW is essentially a spin 1/2 wave propagating on the lattice. Since a spin 1/2 wave obeys the Dirac equation, a DTQW essentially defines discrete equivalents to all quantities appearing in the Dirac equation i.e., an $n$-bein, and thus a metric, and a spin-connection. Once one has a discrete equivalent of the spin-connection, one can compute its curvature (in the sense of gauge theories), which coincides with the Riemann curvature. It is remarkable that classical Markov chains thus provide only a generalization of the Ricci curvature while quantum walks deliver equivalents to all geometrical objects of usual interest, from the metric to the spin-connection and, thus to the full Riemann curvature tensor.

As already mentioned, curvature is often introduced in differential geometry without using spinors. A standard approach is to first define a notion of parallel transport for tensor (including vector and 1-form) fields and then introduce curvature as the natural object which measures how much parallel transport along an infinitesimal closed loop modifies a tensor field. The parallel transport generates a covariant derivative of tensor fields and, thus, a space-time connection, encoded for example in the Christoffel symbols, which can be used to compute the space-time curvature. However, the space-time connection can also be represented by the so-called rotation coefficients, which determine the covariant derivatives of the $n$-bein. Now, this space-time connection can be extended in a canonical way to spinor fields. One can thus define the covariant derivatives of spinor fields and these can be used to compute directly the space-time curvature. This approach presents the advantage of being very close in spirit to Yang–Mills gauge theories: at each point in space-time the fiber is the spinor Hilbert space, the theory is invariant under the local action of various gauge groups ($U(1)$, the Lorentz group, etc.) in this fiber, and one computes for each group the curvature or field strength from the associated covariant spinor derivative.

Let us add a few comments about the mass $\mathcal{M}$. If one focuses on usual physics, only the continuous limit counts and it has been shown in [22,23] that the mass $\mathcal{M}$ then vanishes. Thus, at the continuous limit, the DTQW under consideration describes the continuous dynamics of massless fermions. By definition, the DTQW does not describe known physics outside this limit i.e., in the discrete regime. In particular, we decided to call the matrix $\mathcal{M}$ the mass of the DTQW because the form (30) of the discrete equation resembles the form (6) of the continuous equation and the coefficient in front of $\Psi$ in (6) is $-i$ times the physical mass. However, this does not presuppose that the matrix $\mathcal{M}$, outside the continuous limit, shares any property with physical masses. In particular, the two coefficients $\mathcal{M}_+^-$ and $\mathcal{M}_-^+$ are not necessarily identical outside the continuous limit, where both vanish. In this sense, the matrix $\mathcal{M}$ should be considered as a generalised mass, which becomes a physical mass only at the continuous limit. Note that it is possible to construct DTQWs close to those considered in this article and whose continuous limits describe fermions of non vanishing mass (see, for example, [29]). Let us also recall that some DTQWs not considered in this manuscript have, even in the continuous regime, complex mass terms [23] which thus do not share the properties of physical masses.

Let us now conclude by mentioning possible extensions of this work. One should first address more general DTQWs coupled to arbitrary Yang–Mills fields. The extension to both higher dimensional space-times and higher spins should also prove interesting, starting with walks defined on square lattices, then moving on to more general grids, the ultimate goal being DTQWs on graphs. For example:

what are the necessary graph properties for a DTQW to define a curvature on the graph? Or, how can one use graph geometry to write more efficient quantum algorithms? One should finally extend all these computations to alternate, comparable discrete models such as Lattice Gauge Theories (LGTs) and compare the results with those obtained for DTQWs.

**Funding:** This research received no external funding.

**Acknowledgments:** For R.D.

**Conflicts of Interest:** The author declares no conflict of interest.

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
