# Peer review of "Discrete Geometry from Quantum Walks"

_condensedmatter, doi:10.3390/condmat4020040_

Round 1
Reviewer 1 Report
This paper puts forward the idea of defining concepts from differential geometry and relativistic fields in curved spacetime in connection with the dynamics of discrete quantum walks (QWs). This idea is indeed interesting and deserves further attention. However, I found several crucial aspects that, in my opinion, need to be addressed before I can recommend this paper for publication. Here are my comments:
- The quantum walk defined in Eq. (9) is the same as introduced in Ref. [22], which was shown to reproduce, in the continuum limit, the Dirac equation of a massless Dirac fermion in an arbitrary gravitational field. In fact, the only parameter in the QW, i.e. the angle, is used to simulate the metric (see also the example in Sect. 2.10). I wonder how one can then have enough freedom to make a sensible definition of mass (Sect. 2.3). When the Dirac equation is put under the form Eq. (1), the mass term clearly appears as positive and proportional to the identity, a property that warranties invariance under symmetry transformations. However, the last term in Eq. (30) does not seem to have these properties. In particular, as discussed by the author, this term is not invariant under Lorentz transformations. The proposal made after Eq. (36) addresses this problem, but is this "squared mass" at least positive definite? The same question can be made about the mass obtained in the example, Eq. (81).
- I liked the idea of introducing a curvature from the QW. However, I also think that the corresponding sections need some clarification. Why do we need two different definitions? The "slow" discrete Riemann curvature tensor is claimed to reproduce, in the continuum (up to a constant), the usual Riemann curvature tensor, a property that clearly favors this definition. What does the definition in Sect. 2.6 represent? Why is a reference connection needed? Does this definition depend on the choice of the reference connection? These questions need to be explained.
- Concerning the curvature defined in Sect. 2.7, does the singularity mentioned after Eq. (62) originate from dropping the second (discrete) derivative D_{pp}\Lambda?
- I tried to reproduce Eq. (6), but I obtained different combinations of the n-bein. I noticed that the definition of \gamma^1 differs from [22]. Also, I obtained a different sign on the rhs. Please check.
Only if these questions and comments are properly addressed will I recommend this manuscript for publication.
Some minor points:
- After Eq. (1): The spinor is \Psi. What does it mean that the n-bein coefficients are assumed to be symmetrical? In what sense?
- The reference just before Eq. (1) is missing.
- Just before Eq. (7), it should be mentioned that T is a spin-dependent spatial translation operator.
- The notation used for the matrix in Eq. (80) is confusing, since that matrix corresponds to \sigma_x.
Author Response
See uploaded file.

Reviewer 2 Report
The present manuscript analyzes a class of discrete-time quantum walks and provides a description of the resulting 2D discrete space-time curvature.
The paper provides an interesting analysis on discrete-time quantum walks, and I would thus support publication of the present manuscript in Condensed Matter.
I have only a few comments I would like the author to address:
1) As stated in the conclusions, the author provides a definition of discrete curvature which is calculated in this paper for the system under investigation. Could the author provide a more direct comparison with other definitions (if this is applicable)?
2) The author has also cited two previous papers [22,23] which he coauthored on related topic. The papers are also cited in line 100, page 6, when discussing the gauge change. I would suggest the author to provide a few comments in the introduction discussing the further advances provided by this paper.
3) The are a very few typos which the author should fix:
- line 14, please change "algorithmics" to "algorithms".
- sec. 2.1, first line, please fix the missing reference [?]
Author Response
See uploaded file.

Round 2
Reviewer 1 Report
The author has successfully addressed the questions raised on my previous report. I recommend publication.